

# The impact of obesity on static and proactive balance and gait patterns in sarcopenic older adults: an analytical cross-sectional investigation

Hamza Ferhi[1] and Wael Maktouf[2]

[1] Research Unit (UR17JS01) Sport Performance, Health & Society, Higher Institute of Sport and Physical Education of Ksar Saîd, Tunis, Manouba, Tunisie
[2] Bioengineering, Tissues and Neuroplasticity, UR 7377, Faculty of Medicine, University of Paris Est Créteil, France

Corresponding author
Wael Maktouf,
wael.maktouf@u-pec.fr

## ABSTRACT

**Background:** Obesity is increasingly recognized as a significant factor in the susceptibility of older adults to falls and related injuries. While existing literature has established a connection between obesity and reduced postural stability during stationary stances, the direct implications of obesity on walking dynamics, particularly among the older adults with sarcopenia, are not yet comprehensively understood.

**Objective:** Firstly, to investigate the influence of obesity on steady-state and proactive balance, as well as gait characteristics, among older adults with sarcopenic obesity (SO); and secondly, to unearth correlations between anthropometric characteristics and balance and gait parameters in the same demographic.

**Methods:** A cohort of 42 participants was categorized into control (CG; $n = 22$; age = 81.1 ± 4.0 years; BMI = 24.9 ± 0.6 kg/m²) and sarcopenic obese (SOG; $n = 20$; age = 77.7 ± 2.9 years; BMI = 34.5 ± 3.2 kg/m²) groups based on body mass index (BMI, kg/m²). Participants were assessed for anthropometric data, body mass, fat and lean body mass percentages (%), and BMI. Steady-state balance was gauged using the Romberg Test (ROM). Proactive balance evaluations employed the Functional Reach (FRT) and Timed Up and Go (TUG) tests. The 10-m walking test elucidated spatiotemporal gait metrics, including cadence, speed, stride length, stride time, and specific bilateral spatiotemporal components (stance, swing, 1st and 2nd double support, and single support phases) expressed as percentages of the gait cycle.

**Results:** The time taken to complete the TUG and ROM tests was significantly shorter in the CG compared to the SOG ($p < 0.05$). In contrast, the FRT revealed a shorter distance achieved in the SOG compared to the CG ($p < 0.05$). The CG exhibited a higher gait speed compared to the SOG ($p < 0.05$), with shorter stride and step lengths observed in the SOG compared to the CG ($p < 0.05$). Regarding gait cycle phases, the support phase was longer, and the swing phase was shorter in the SOG compared to the CG group ($p < 0.05$). LBM (%) showed the strongest positive correlation with the ROM ($r = 0.77$, $p < 0.001$), gait speed ($r = 0.85$, $p < 0.001$), TUG ($r = -0.80$, $p < 0.001$) and FRT ($r = 0.74$, $p < 0.001$).

**Conclusion:** Obesity induces added complexities for older adults with sarcopenia, particularly during the regulation of steady-state and proactive balance and gait.

The percentage of lean body mass has emerged as a crucial determinant, highlighting a significant impact of reduced muscle mass on the observed alterations in static postural control and gait among older adults with SO.

## INTRODUCTION

As the global senior population expands, so too does the prevalence of obesity—a health concern intensified by the simultaneous increase in adipose tissue and decrease in muscle strength and mass (*Zamboni, Rubele & Rossi, 2019*; *Colleluori & Villareal, 2021*; *Li et al., 2022*). This phenomenon gives rise to a condition known as sarcopenic obesity (SO). SO notably elevates the risk of various health challenges, particularly functional impairments and heightened injury vulnerability (*Kopelman, 2000*; *Bray, 2004*). Particularly worrisome is SO's potential impact on routine activities, such as walking (*Liao et al., 2022*). Maintaining the ability to walk is essential in older adults with SO, as it plays a vital role in both disease prevention and autonomy preservation (*Malatesta et al., 2009*).

Several factors, ranging from neural and hormonal shifts to environmental changes, accompany aging, leading to rapid muscle deterioration (*Vandervoort, 2002*; *Frontera, 2017*). Underlying causes for this decline include reduced physical activity (*Cunningham et al., 2020*), hormonal imbalances (*van den Beld et al., 2018*), increased inflammatory response (*Campisi et al., 2020*), and neuronal losses in the central nervous system (*Li, Xiong & Mei, 2018*). Consequently, these combined effects jeopardize basic functional capabilities, destabilize posture, and inhibit walking proficiency (*Chen & Chou, 2022*; *Beck Jepsen et al., 2022*), enhancing fall risk (*Nascimento et al., 2022*; *Macie, Matson & Schinkel-Ivy, 2023*). Within this landscape, sarcopenia manifests as a multifaceted syndrome affecting older adults' walking stability, leading to noticeable reductions in gait speed, balance, and stride length (*Doherty, 2003*; *Frontera, 2017*).

The adverse effects of adult obesity on balance during static stances are well-acknowledged (*Hue et al., 2007*; *Handrigan et al., 2010*; *Dutil et al., 2013*; *Maktouf et al., 2018*). Factors such as increased ankle muscle activity in obese individuals can limit postural control, impacting dynamic performances like walking (*Tucker et al., 2008*; *Allum et al., 2002*; *Wu, 2008*). Concomitantly, researchers have identified gait variations among obese adults, with these individuals often exhibiting slower walking speeds, altered stride lengths, and modifications in the gait cycle (*De Souza et al., 2005*; *Peyrot et al., 2009*; *Ko, Stenholm & Ferrucci, 2010*). While such studies have shed light on obesity's repercussions on gait in general populations, the specific effects on older sarcopenic individuals remain a contested topic. Although some researchers assert that obesity doesn't necessarily hinder mobility (*Sallinen et al., 2011*; *Meng et al., 2014*; *Pereira et al., 2017*; *Santos et al., 2017*; *Gonzalez, Gates & Rosenblatt, 2020*), others identify a discernible negative influence on gait and overall physical functionality (*Carneiro et al., 2012*; *Dutil et al., 2013*; *Maktouf et al., 2018*, *2019*, *2020*; *Kong, Won & Kim, 2020*; *Liao et al., 2022*). This inconsistency may stem

from a lack of proper categorization and in-depth exploration of SO in many studies. These disparities in findings might be attributed to the oversight of sarcopenia identification in older study subjects. Few of these studies categorize participants as sarcopenic, let alone delve into the intricacies of SO. Nevertheless, it is imperative to undertake more comprehensive research to clarify the influence of obesity on both steady-state and proactive balance, as well as specific gait parameters in sarcopenic older adults. Such insights are indispensable for crafting targeted therapeutic interventions and enhancing mobility in elderly individuals grappling with SO.

Research has illuminated the strong ties between anthropometric features and balance in obese individuals (*Hue et al., 2007*; *Cruz-Gómez et al., 2011*; *Handrigan et al., 2017*). Yet, when it comes to SO in older adults, there is a knowledge gap. Discerning whether reduced muscle mass or heightened body fat predominantly impacts functional capacities in SO-affected elderly remains a pivotal question. Upcoming research should focus on dissecting the influence of specific anthropometric indices on balance and walking, given the augmented fall risks in this group. Such research will elucidate strategies to enhance balance and gait in this demographic, ultimately mitigating the risk of falls.

This study pursues dual objectives: firstly, to investigate the impact of obesity on steady-state and proactive balance and gait parameters in older adults with SO; and secondly, to unearth correlations between anthropometric parameters and balance and gait traits in the same demographic.

## MATERIALS AND METHODS

This study was structured using an analytical cross-sectional design (Fig. 1) and conducted over 4 months. The entire experimental phase was divided into three critical stages. Initially, a recruitment phase took place, spanning a period of 4 weeks. This was followed by a screening duration that lasted between 1 to 3 weeks. Finally, an intensive experimental testing period of 9 weeks was executed. During the testing phase, participants underwent a comprehensive experimental protocol, which spanned across 2 h. This protocol was composed of five pivotal assessments. Firstly, participants were presented with questionnaires aimed at gauging their current health status. Subsequently, they underwent anthropometric measurements, and their static steady-state balance was meticulously evaluated. The fourth assessment revolved around proactive balance, and the sequence culminated with the 10-m walking test.

### Participants

The estimation of our study's sample size was rigorously determined using G*Power (version 3.1.9.4), with parameters set to control Type I error at alpha = 0.05 and Type II error at beta = 0.60. With an anticipated moderate effect size of r = 0.35, this power analysis indicated a necessity for at least 40 participants. Embarking on recruitment between January and April 2022, 72 volunteers were initially garnered from various regional obesity care centers. Strict adherence to our pre-established inclusion and exclusion criteria meant that of these, only 45 candidates qualified. However, due to challenges related to adherence to our study's regimen, we witnessed a drop out of three individuals, culminating in a final

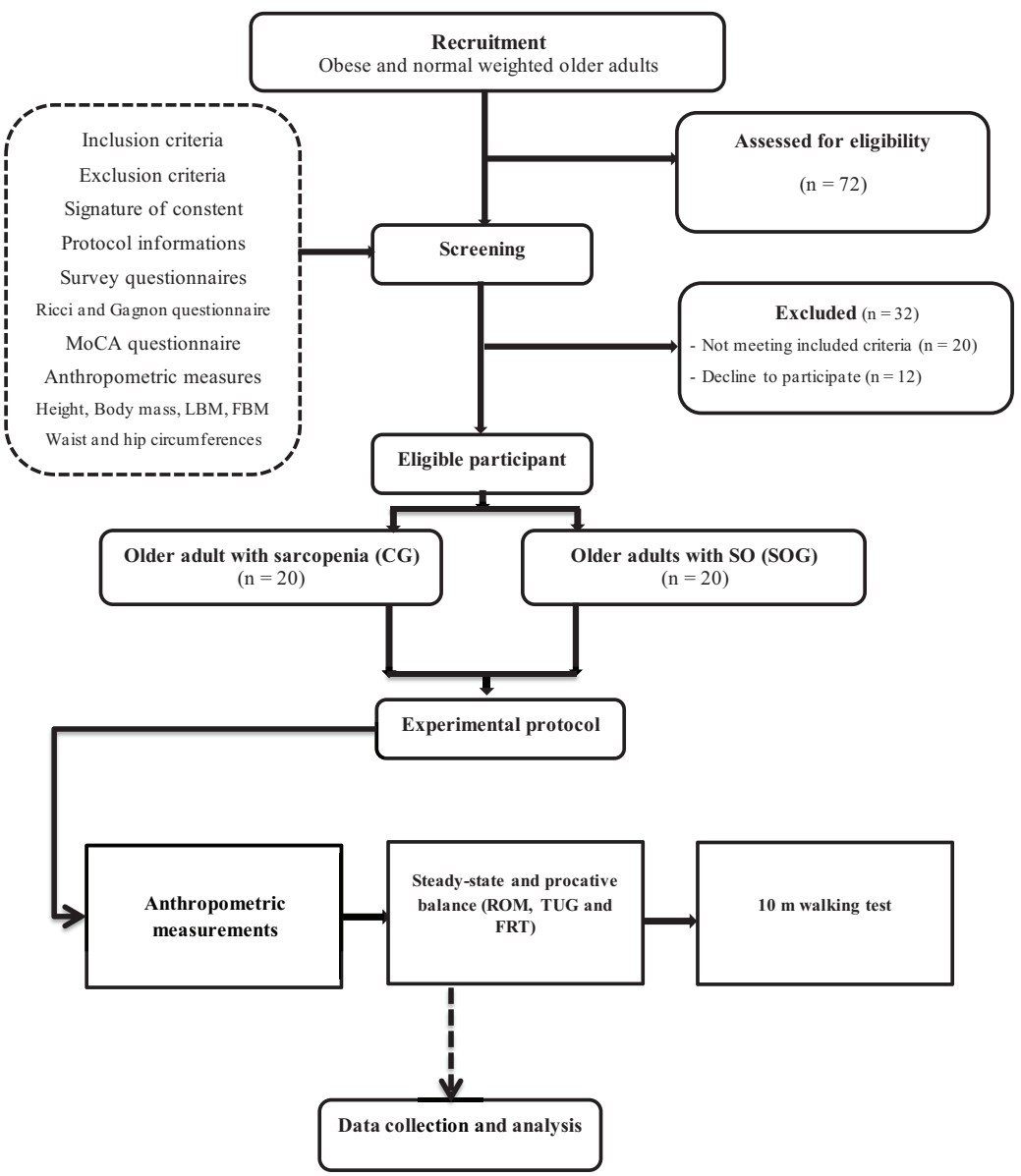

**Figure 1 Experimental procedure design.**

sample size of 42 participants. Based on their body mass index (BMI, kg/m²), they were subsequently allocated into two distinct groups: the control group (CG; $n$ = 22; M/F = 12/10; age = 81.1 ± 4.0 years; BMI = 24.9 ± 0.6 kg/m²) and the sarcopenic obese group (SOG; $n$ = 20; M/F = 12/8; age = 77.7 ± 2.9 years; BMI = 34.5 ± 3.2 kg/m²).

To qualify, participants had to demonstrate a handgrip force less than 17 N, a gait speed under 1.0 m/s, be aged over 65 years, maintain the capacity for verbal communication with our team, and sustain physical independence. Individuals presenting with conditions such as severe neurological or cognitive impairments, significant cardiovascular diseases, major musculoskeletal deformities or injuries, or other chronic diseases, as well as those on medications potentially affecting our assessments or having a Montreal Cognitive

Assessment (MoCA) score below 26, were systematically excluded. As a preliminary step, we had each potential participant fill out a detailed questionnaire, shedding light on aspects like the history of falls, medication usage, physical independence, and frailty levels. Their responses were meticulously examined by the medical professionals stationed at the recruitment centers. Questionnaires like the Ricci and Gagnon scales (*Zulfiqar et al., 2022*) and the MoCA test (*Smith, Gildeh & Holmes, 2007*) were employed to evaluate physical activity and cognitive statuses, respectively. The onset of the study was marked by each participant signing an informed consent form, reflective of their complete understanding and voluntary participation. The study was conducted in accordance with the Declaration of Helsinki and approved by the Ethics Committee of South Ethics Committee for the Protection Persons (C.P.P. SOUTH /No. 0477/2022, 22 February 2022). Throughout the research, a steadfast commitment was maintained towards ensuring the confidentiality of participant data, with all analyses being conducted in an anonymous and aggregated manner, upholding the highest standards of research ethics.

## Evaluation protocol

To ensure uniformity and eliminate variance, all these tests were administered in a specialized clinical examination room by a singular assessor. Additionally, to maintain clarity and coherence, participants were given standardized verbal directives concerning the procedural aspects of the tests.

### Anthropometric measurements

Anthropometric measurements were conducted, encompassing height, waist circumference, and hip circumference using a tape measure. Body mass (BM), percentage of body fat mass (FBM, %) and percentage of lean body mass (LBM, %) were assessed using an impedance meter, a sophisticated device that gauges electrical impedance as it passes through the body. These meters typically consist of electrodes that are placed on or in contact with the feet. A small, safe electrical current is passed through the body *via* these electrodes. The impedance meter then measures how the electrical current is impeded as it travels through various tissues, including fat and lean muscle. From these measurements, the fat body mass (FBM) and LBM were calculated using the equations from *Gartner et al. (2004)*: FBM = body fat (%) × body mass; and LBM = LBM (%) × body mass.

### Evaluation of static steady-state balance

The evaluation of static steady-state balance was performed utilizing the Romberg Test (ROM) (*Gschwind et al., 2013*). Participants were directed to maintain an upright position for 30 s without wearing shoes. They were instructed to keep their feet close together and extend their arms fully in front of their bodies, with palms facing upwards, while keeping their eyes closed. If participants opened their eyes, made arm or foot movements to regain stability, or needed assistance from the operator, the test was terminated. Each participant completed three trials, with a 1-min rest period between each trial, and the best-recorded result was noted as the standing time in seconds.

*Evaluation of proactive balance*

To evaluate proactive balance, two tests were employed: the Functional Reach Test (FRT) and the Timed Up and Go Test (TUG) (*Gafner et al., 2021*; *Ortega-Bastidas et al., 2023*). During the FRT, participants were instructed to raise their dominant arm and create a fist. Upon hearing an auditory cue, they extended their arm along an adjustable tape measure, reaching as far as possible. The FRT consisted of three 12-s trials, and if any step or contact with the tape measure occurred, the trial was terminated. The maximal reach distance was measured and recorded in centimeters. For the TUG test, participants were instructed to sit on a chair with a height of 46 cm and position their arms on the armrests. They were then asked to rise from the chair, walk 3 m at their usual walking pace, turn around, and return to a seated position. Two test trials were conducted, and the best time achieved in seconds was recorded as the outcome measure.

*The 10-meter walking test*

Participants were instructed to walk along a 20-m corridor to determine their maximum gait speed (m/s). To ensure consistent measurements, only the speed between the 5th and 15th meters was considered, excluding the acceleration and deceleration phases. For gait analysis, a wireless inertial sensor system (BTS Bioengineering S.p.A., Milan, Italy) was utilized. The sensors were attached using a semi-elastic belt positioned at the level of the fifth lumbar vertebra (L5) and the first two sacral vertebrae (S1-S2). Data were collected at a frequency of 100 Hz and transmitted wirelessly *via* Bluetooth 3.0 to a computer. A dedicated software 3.0 (BTS G-Studio) was employed to process the data and calculate spatiotemporal gait parameters. The recorded spatiotemporal parameters included cadence (strides/min), speed (m/s), stride length (cm), and stride time (s). Bilateral spatiotemporal parameters, expressed as a percentage of the gait cycle, stance phase (% of the gait cycle), swing phase (% of the gait cycle), 1st double support (% of the gait cycle), single support (% of the gait cycle), and 2nd double support (% of the gait cycle) phases.

## Statistical analysis

The statistical analysis was carried out using Jamovi software (Version 2.3, Sydney, Australia). Prior to analysis, normality and homogeneity of variance assumptions were checked using the Shapiro-Wilk and Levene tests, respectively. All parameters met the assumptions of normality and homogeneity of variance. To compare differences between groups, independent t-tests were performed for all parameters, with obesity as the grouping variable. Additionally, Spearman correlation analysis was utilized to examine associations between anthropometric measures and gait/balance parameters. Mean values with their respective standard deviations were reported, and the significance level was set at $p < 0.05$.

## RESULTS

### Anthropometric measurements

Table 1 presents the anthropometric measurements of the CG and the SOG groups. The statistical analysis revealed that FBM, body fat, body mass, and BMI were higher in

**Table 1  Anthropometric characteristics of groups.**

| | | | | 95% Confidence interval | | | | |
|---|---|---|---|---|---|---|---|---|
| | Group | N | Mean | Lower limit | Upper limit | Standard-deviation | Minimum | Maximum |
| Age (years) | SOG | 20 | 77.7 | 76.33 | 79.1 | 2.954 | 72.80 | 83.0 |
| | CG | 22 | 81.1 | 79.35 | 82.9 | 4.020 | 74.90 | 88.0 |
| Height (cm) | SOG | 20 | 162.9 | 159.92 | 165.8 | 6.294 | 152.60 | 173.5 |
| | CG | 22 | 166.0 | 162.69 | 169.4 | 7.538 | 155.80 | 177.2 |
| Body mass (kg) | SOG | 20 | 91.0*** | 89.14 | 92.8 | 3.923 | 84.20 | 96.8 |
| | CG | 22 | 68.7 | 66.28 | 71.2 | 5.542 | 60.40 | 79.0 |
| BMI (kg/m²) | SOG | 20 | 34.5*** | 32.94 | 36.0 | 3.247 | 30.11 | 41.3 |
| | CG | 22 | 24.9 | 24.64 | 25.2 | 0.635 | 23.68 | 26.3 |
| Body fat (%) | SOG | 20 | 35.0*** | 32.03 | 38.0 | 6.340 | 24.30 | 45.6 |
| | CG | 22 | 17.7 | 16.84 | 18.6 | 1.982 | 14.30 | 20.9 |
| LBM (%) | SOG | 20 | 65.0*** | 62.22 | 67.78 | 1.982 | 54.4 | 75.7 |
| | CG | 22 | 82.3 | 81.45 | 83.11 | 6.340 | 79.1 | 85.7 |
| FBM (kg) | SOG | 20 | 32.0*** | 28.78 | 35.3 | 6.932 | 21.75 | 44.1 |
| | CG | 22 | 12.2 | 11.52 | 12.8 | 1.425 | 9.47 | 15.6 |
| LBM (kg) | SOG | 20 | 59.0 | 57.01 | 60.9 | 4.153 | 52.66 | 67.8 |
| | CG | 22 | 56.6 | 54.31 | 58.9 | 5.146 | 49.29 | 65.8 |
| Handgrip force (N) | SOG | 20 | 11.7 | 9.96 | 13.4 | 3.651 | 5.20 | 18.8 |
| | CG | 22 | 12.7 | 10.75 | 14.6 | 4.291 | 5.20 | 20.8 |

Notes:
SOG, older adults with sarcopenic obesity; CG, control group.
*** $p < 0.001$.

SOG compared to CG ($p < 0.001$). However, percentage of LBM was lower in SOG than CG ($p < 0.001$). No significant effect was revealed on age, height, LBM, and handgrip force.

## Steady-state and proactive balance

Table 2 presents the results of steady-state and proactive balance evaluations for both the CG and SOG groups. Statistical analysis indicated that the distance achieved in the FRT was significantly shorter in the SOG group compared to the CG ($p = 0.036$, d = 0.67). Additionally, the time taken to complete the TUG test was significantly shorter in the CG compared to the SOG group ($p = 0.006$, d = 0.9). While the standing time in the ROM test was significantly longer in the CG compared to the SOG group ($p = 0.002$, d = −1.01).

## Gait parameters

Table 3 presents the gait parameters for both the CG and SOG groups. The statistical analysis revealed significant differences in gait parameters between the two groups. Specifically, the gait speed was lower in the SOG compared to CG ($p < 0.001$, d = −1.63). The stride length and step length were all significantly lower in the SOG group compared to the CG ($p < 0.001$, d = −6.67; $p < 0.001$, d = −6.67, respectively). In terms of gait cycle phases, the support phase was significantly longer, while the swing phase was significantly

**Table 2 Results of steady-state and poactive balance tests across groups.**

| | Group | N | Mean | Standard error | 95% Confidence interval Lower limit | 95% Confidence interval Upper limit | Standard-deviation | Minimum | Maximum |
|---|---|---|---|---|---|---|---|---|---|
| FRT (cm) | SOG | 20 | 8.74* | 0.499 | 7.70 | 9.79 | 2.23 | 5.30 | 12.8 |
| | CG | 22 | 10.31 | 0.518 | 9.24 | 11.39 | 2.43 | 5.90 | 14.2 |
| TUG (s) | SOG | 20 | 12.96** | 0.300 | 12.33 | 13.58 | 1.34 | 10.70 | 14.7 |
| | CG | 22 | 11.60 | 0.349 | 10.88 | 12.33 | 1.64 | 8.10 | 13.9 |
| Romberg test (s) | SOG | 20 | 8.74** | 0.520 | 7.65 | 9.83 | 2.32 | 5.30 | 13.4 |
| | CG | 22 | 11.11 | 0.505 | 10.06 | 12.16 | 2.37 | 5.90 | 15.0 |

Notes:
SOG, older adults with sarcopenic obesity; CG, control group.
* $p < 0.05$.
** $p < 0.01$.

**Table 3 Gait parameters across groups.**

| | Group | N | Mean | Standard error | 95% Confidence interval Lower limit | 95% Confidence interval Upper limit | Standard deviation | Minimum | Maximum |
|---|---|---|---|---|---|---|---|---|---|
| Gait speed (m/s) | SOG | 20 | 0.700*** | 0.0242 | 0.649 | 0.750 | 0.108 | 0.490 | 0.900 |
| | CG | 22 | 0.908 | 0.0306 | 0.845 | 0.972 | 0.144 | 0.630 | 1.200 |
| Step length (cm) | SOG | 20 | 24.195*** | 0.2213 | 23.732 | 24.658 | 0.990 | 22.700 | 26.200 |
| | CG | 22 | 33.973 | 0.3815 | 33.179 | 34.766 | 1.789 | 30.300 | 36.400 |
| Stride length (cm) | SOG | 20 | 48.390*** | 0.4426 | 47.464 | 49.316 | 1.979 | 45.400 | 52.400 |
| | CG | 22 | 67.945 | 0.7629 | 66.359 | 69.532 | 3.578 | 60.600 | 72.800 |
| Support phase (%) | SOG | 20 | 74.670*** | 0.2397 | 74.168 | 75.172 | 1.072 | 71.200 | 76.100 |
| | CG | 22 | 68.618 | 0.2598 | 68.078 | 69.159 | 1.219 | 66.500 | 70.700 |
| 1st double support phase (%) | SOG | 20 | 26.460*** | 0.2899 | 25.853 | 27.067 | 1.296 | 23.900 | 29.400 |
| | CG | 22 | 22.450 | 0.6578 | 21.082 | 23.818 | 3.086 | 19.700 | 31.100 |
| Single support phase (%) | SOG | 20 | 23.745*** | 0.2528 | 23.216 | 24.274 | 1.131 | 21.800 | 25.700 |
| | CG | 22 | 26.777 | 0.1918 | 26.378 | 27.176 | 0.900 | 25.300 | 28.400 |
| 2nd double support phase (%) | SOG | 20 | 24.465*** | 0.4569 | 23.509 | 25.421 | 2.043 | 21.000 | 28.200 |
| | CG | 22 | 19.391 | 0.6695 | 17.999 | 20.783 | 3.140 | 11.000 | 23.100 |
| Swing phase (%) | SOG | 20 | 25.330*** | 0.2397 | 24.828 | 25.832 | 1.072 | 23.900 | 28.800 |
| | CG | 22 | 31.382 | 0.2598 | 30.841 | 31.922 | 1.219 | 29.300 | 33.500 |

Notes:
SOG, older adults with sarcopenic obesity; CG, control group.
*** $p < 0.001$.

shorter in the SOG group compared to the CG ($p < 0.001$, d = 5.26; $p < 0.001$, d = $-5.26$, respectively).

## Correlation analysis

The results from the data highlight significant correlations between various anthropometric measures and different balance and gait parameters in SOG (Table 4). Body weight exhibited a significant negative correlation with the ROM and the FRT with correlation values of $-0.51$ ($p = 0.02$) and $-0.51$ ($p = 0.022$), respectively. Interestingly, it

**Table 4 Correlation analysis between anthropometric measurements and gait and balance parameters.**

| | | | ROM (s) | FRT (cm) | TUG (s) | Gait speed (m/s) |
|---|---|---|---|---|---|---|
| **Body weight (kg)** | SOG | r | −0.51 | −0.51 | 0.61 | −0.62 |
| | | p | 0.02 | 0.022 | 0.004 | 0.004 |
| | CG | r | −0.13 | −0.11 | −0.37 | 0.36 |
| | | p | 0.571 | 0.629 | 0.088 | 0.368 |
| **BMI (kg/m²)** | SOG | r | −0.54 | −0.51 | 0.64 | −0.64 |
| | | p | 0.014 | 0.022 | 0.002 | 0.002 |
| | CG | r | −0.13 | −0.11 | −0.37 | 0.2 |
| | | p | 0.571 | 0.629 | 0.088 | 0.369 |
| **Lean body mass (%)** | SOG | r | 0.77 | 0.74 | −0.80 | 0.85 |
| | | p | <0.001 | <0.001 | <0.001 | <0.001 |
| | CG | r | 0.12 | 0.15 | 0.18 | −0.12 |
| | | p | 0.609 | 0.501 | 0.415 | 0.599 |

Note:
   r, coefficient of correlation; LBM, Lean body mass; FBM, Fat body mass; FRT, functional reach test; TUG, Time up and go test; ROM, Romberg test; BMI, body mass index.

showed a positive correlation with the TUG test ($r = 0.61$, $p = 0.004$) but a negative correlation with gait speed ($r = -0.62$, $p = 0.004$). BMI showed a significant negative correlation with the ROM ($r = -0.54$, $p = 0.014$) and FRT ($r = -0.51$, $p = 0.022$) while being positively correlated with the TUG test ($r = 0.64$, $p = 0.002$) and negatively with gait speed ($r = -0.64$, $p = 0.002$). The percentage of LBM revealed a notable positive correlation with the ROM ($r = 0.77$, $p < 0.001$) and the FRT ($r = 0.74$, $p < 0.001$). Additionally, it exhibited a negative correlation with the TUG test ($r = 0.80$, $p < 0.001$) but was positively correlated with gait speed ($r = 0.85$, $p < 0.001$) (Fig. 2). Finally, no significant correlations were observed between anthropometric parameters and balance and gait measures (Fig. 2).

## DISCUSSION

The core focus of this investigation was to unravel the effects of obesity on balance and walking characteristics in elderly individuals affected by SO. Additionally, we sought to understand the relationship between various anthropometric attributes and corresponding gait and balance measurements. Our data underscores that obesity negatively impacts both steady-state and proactive balance attributes. Furthermore, our analysis of walking dynamics reveals that obesity significantly alters various walking parameters, such as walking speed, step distance, and overall stride measurements. The percentage of lean body mass demonstrates a strong correlation with balance and gait parameters in older adults with SO, but not in those without obesity. Correlations become notably stronger when LBM (%) falls below 75% or when FBM (%) exceeds 25%. From a clinical perspective, it is conceivable to consider these values as an alert threshold, indicating a high risk of balance and walking alterations, thus potentially increasing the risk of falls. These findings serve as valuable pointers for the development of tailored physical activity regimens for individuals diagnosed with SO.

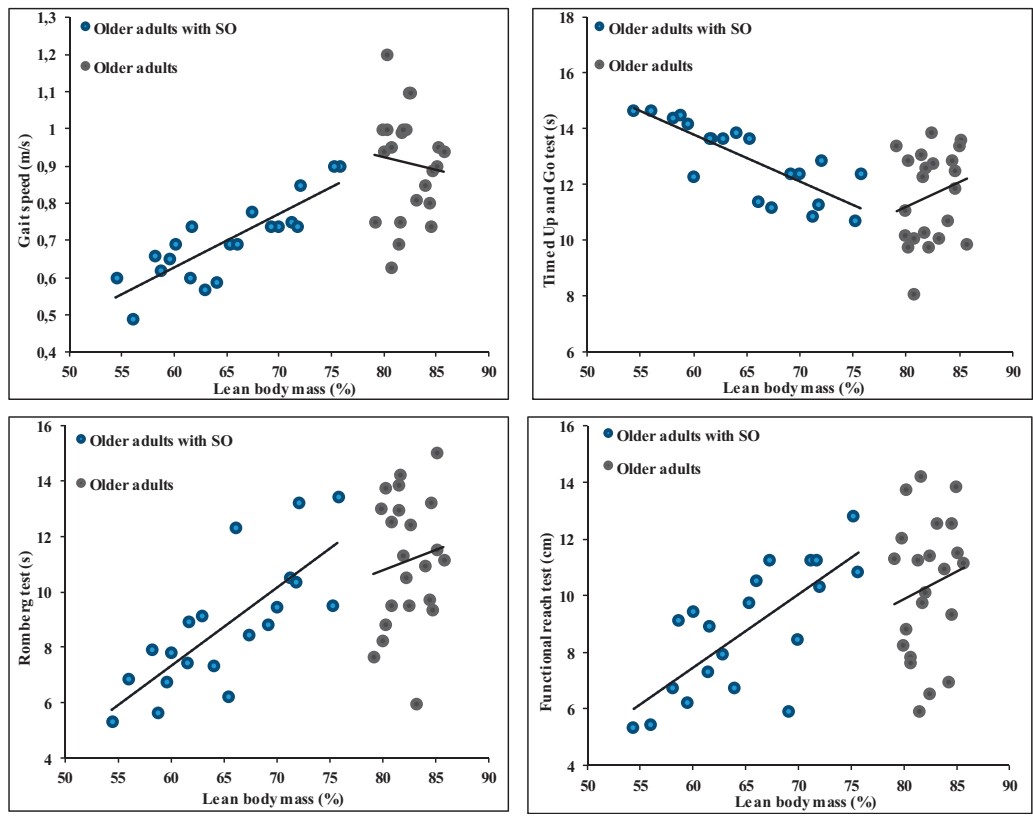

**Figure 2** **Relationships between percentage of lean body mass and balance and gait parameters in older adults without and with sarcopenic obesity.** No significant correlations between LBM (%) and gait and balance parameters for older adults without SO. For older adults with SO, significant correlations were observed between LBM (%) and gait speed (r = 0.85), Romberg test (r = 0.77), functional Reach Test (r = 0.74) and Timed Up and Go test (r = −0.80).

## Effect of obesity on the steady-state and proactive balance

Our analysis reveals a definitive influence of obesity on the balance of older adults with sarcopenia. When considering the steady-state balance, as evaluated through the ROM test, there was a noticeable 20% lower performance among the older SO group. In the context of proactive balance evaluations, our data from both FRT and TUG tests revealed a 14% lower FRT score and a 12.5% higher TUG timing. Such results resonate with existing research, particularly those focused on older adults not classified as sarcopenic (*Carneiro et al., 2012*; *Dutil et al., 2013*; *Maktouf et al., 2018, 2019, 2020*; *Kong, Won & Kim, 2020*; *Liao et al., 2022*). Our findings emphasize the compounded adversities posed by obesity in conjunction with the inherent challenges of aging, accentuating their combined toll on functional capabilities (*Dutil et al., 2013*; *Handrigan et al., 2017*; *Maktouf et al., 2018, 2020*). The amalgamation of obesity and aging not only escalates the scope of functional deficits but may also increase the predisposition towards accidents like falls and related injuries (*Baumgartner et al., 2004*; *Handrigan et al., 2017*; *Li et al., 2022*).

To comprehend the amplified risk factors for falls among the obese populace, scholarly investigations put forth three primary conjectures. Initially, the incessant strain from bearing excess weight often results in decreased sensitivity in the foot soles, attributable to

overstimulation of the plantar mechanoreceptors (*Handrigan et al., 2012b*; *Wu & Madigan, 2014*). The next rationale hinges on the biomechanical challenges of sustaining augmented body mass, especially when a sizable chunk of this mass is positioned away from the pivotal axis (using the ankle joint's inverted pendulum paradigm). This configuration necessitates an increased muscular torque to counterbalance the heightened gravitational pull and maintain verticality (*Corbeil et al., 2001*; *Simoneau & Teasdale, 2015*). A third proposition emphasizes cognitive demands; these might impose additional challenges in balance management for obese subjects (*Mignardot et al., 2010*). The implications suggested by these conjectures seem to be magnified in the presence of sarcopenia, a muscle degeneration condition. This condition results in sequential changes in the neuromuscular, proprioceptive, and visual apparatus, culminating in a compromise in balance and posture. The intertwined nature of obesity and sarcopenia in the elderly thus significantly impinges upon their balance and postural stability.

## Effect of obesity on gait

In pioneering the evaluation of gait's spatiotemporal parameters in elderly sarcopenic individuals, our study ventured beyond the often-singular focus on gait speed, a characteristic approach in preceding research. However, existing literature on SO and gait exhibits mixed findings. While *Meng et al. (2014)* and *Sallinen et al. (2011)* found minimal effects of SO on gait speed and mobility for those over 80, *Kong, Won & Kim (2020)* and *Stenholm et al. (2009)* associated SO with substantial functional declines.

Corroborating with the latter group, our findings illustrated a decline in gait speed in older adults with SO, characterized by shortened step length. These individuals also exhibited a high support phase duration during walking, characterized by a prolonged double support phase. Interestingly, research on older obese individuals without sarcopenia reveals similar gait dynamics. The lengthened double support phases might be a strategic adaptation to limit postural instability. This is buttressed by postural control theories which posit enhancing dynamic joint stability to counter obesity-induced imbalances (*McGraw et al., 2000*; *Malatesta et al., 2009*).

Evidence suggests obese individuals experience broader foot contact areas and heightened pressure during postural tasks, potentially impacting the feedback from plantar mechanoreceptors (*Hue et al., 2007*; *Wu & Madigan, 2014*). Moreover, obesity might lead to extended double support phases to absorb and propel excessive body mass (*Ko, Stenholm & Ferrucci, 2010*). *Maktouf et al. (2020)* even suggested that body mass plays a substantial role in influencing muscle activity during walking, hinting at the amplified energy expenditure in obese older adults.

In essence, while the observed gait alterations among SO individuals seem adaptive, optimizing their mobility within their strength constraints, they are not without risks. Given the inherent muscular vulnerabilities in SO (*Tomlinson et al., 2016*), an over-reliance on these adaptations may escalate fall risks (*Handrigan et al., 2017*). This juxtaposition underscores the delicate balance that these individuals navigate daily, balancing between mobility optimization and potential hazards.

## The interrelation of body metrics with gait and balance parameters

The novelty of our study lies in its nuanced exploration of anthropometric parameters' influence, especially body weight, BMI and LBM (%) on gait and balance among older adults with SO. Our analysis underscored LBM (%) as a paramount factor, establishing a significant correlation between muscle mass and both postural control and walking competence in older adults with SO. Whereas studies like that by *Hue et al. (2007)* emphasized the role of overall body mass in gait and balance among obese adults, our investigation extends this understanding specifically to the SO subset. The intricate correlations discovered between body metrics and gait and balance assessments suggest that while factors such as body weight and BMI play roles, it is undeniably LBM (%) that exerts the most substantial influence, especially for those with SO. This denotes the distinct difference in body composition and its functional implications between the general obese population and older adults with SO.

Literature reveals that obesity can modify plantar feedback due to enlarged foot contact regions and heightened pressure, potentially undermining balance *via* decreased signals from plantar mechanoreceptors (*Birtane & Tuna, 2004*; *Bensmaïa, Leung & Johnson, 2005*). Additionally, excessive abdominal mass in the obese might demand increased ankle torque for balance, introducing more motor function variability and potentially affecting stability (*Handrigan et al., 2012a*). However, when delving deeper, the salient influence of LBM (%) is perhaps attributed to the muscle decline characteristic of sarcopenia (*Tomlinson et al., 2016*). At their core, muscles exert force fundamental for myriad gait and balance functions (*Cattagni et al., 2014*). In the context of walking, muscles attenuate shock during the first double support phase and later furnish propulsion in the second. Beyond their biomechanical significance, muscle power directly correlates with and predicts balance (*Gimmon et al., 2015*; *Maslivec et al., 2018*). A diminished LBM (%), indicating a decline in muscle mass and consequent force production capability, inherently hampers an individual's gait and balance. As indicated previously, results indicated strong correlations strong correlations between gait and balance parameters with LBM (%) more than BMI and BM. Notably, these correlations were absent in older adults without SO. Moreover, our findings indicate the presence of a threshold effect, where correlations become notably stronger when LBM (%) drops below 75% or when FBM (%) surpasses 25%. From a clinical perspective, it is conceivable to suggest that these values could be considered as an alert threshold, indicating a high risk of balance and walking alterations and consequently an increased risk of falls. Although these results need further confirmation through prospective studies, it is worth noting that it may be relevant to recommend strength training for individuals identified below this threshold to modify their body composition, favoring an increase in muscle mass and strength.

Previous studies have demonstrated that handgrip force serves as a reliable predictor of various negative outcomes, including falls, postsurgical complications, and future disability (*Rijk et al., 2016*; *Benton, Spicher & Silva-Smith, 2022*). However, in our study involving older adults with SO, we did not observe any significant correlation between handgrip force and balance and gait parameters. One possibility is that force production capacities in

the upper limbs of older adults with SO may not necessarily reflect force production in the lower limbs, as observed in older adults without obesity (*Benton, Spicher & Silva-Smith, 2022*). Additionally, it is possible that, below a certain threshold, as seen in our study group where LBM (%) dropped below 75%, there is no correlation between muscle mass and force production. This may be due to neuromuscular system alterations exacerbated by obesity in this particular population (*Erskine et al., 2017*). These intriguing possibilities warrant further investigation to better understand the complex relationship between handgrip force, muscle mass, and neuromuscular function in older adults with SO and how it differs from their counterparts without obesity.

### Limitations and perspectives

Our study's primary strength lies in its unique focus on older adults diagnosed with SO, offering a fresh perspective on the relationship between body mass and lean body mass on gait and balance. However, the study's relatively small cohort of 42 subjects limits the broad generalizability of our findings, especially considering the inherent variability and heterogeneity commonly found in older populations. The use of impedance-meters for deducing LBM and FBM might introduce some imprecision, with tools like dual-energy X-ray absorptiometry offering potentially more accurate insights. Our results highlight the importance of interventions for individuals with SO to focus on enhancing both the quality and quantity of lean muscle mass, while also addressing overall body mass reduction. Programs that solely emphasize weight loss might inadvertently diminish lean muscle mass in tandem with fat mass, potentially exacerbating conditions in sarcopenic subjects due to muscle's pivotal role in ensuring stability and functional mobility. Further research employing more sophisticated tools and larger samples is vital to derive comprehensive intervention strategies for this demographic.

## CONCLUSIONS

Our study accentuates the multifaceted challenges faced by older adults with SO, especially when modulating steady-state and proactive balance, as well as gait attributes. LBM emerges as the keystone, showcasing a marked association between muscle mass and both postural stability and ambulatory proficiency among older adults with SO. Additionally, while body fat also exhibited correlations with gait and balance parameters, its influence was discernible but less pronounced than that of LBM. For interventions to be optimally effective, the strategy should be dual-pronged: curbing excessive body fat while simultaneously amplifying the volume and vitality of muscle mass. Adopting such a comprehensive approach is pivotal for enhancing balance and gait capabilities within older adults with SO.

## ACKNOWLEDGEMENTS

We wish to extend our heartfelt gratitude to the medical personnel and leadership of the institutions that graciously allowed us to recruit participants for this study. Their unwavering support and assistance have been instrumental in the progress and success of our research. Additionally, our sincere appreciation goes out to the participants who

generously took part in our study. Their enthusiasm and collaboration throughout the data-gathering phase were paramount, and we remain deeply thankful for their invaluable contribution.

## ADDITIONAL INFORMATION AND DECLARATIONS.

### Funding
The authors received no funding for this work.

### Competing Interests
The authors declare that they have no competing interests.

### Author Contributions
- Hamza Ferhi conceived and designed the experiments, performed the experiments, prepared figures and/or tables, and approved the final draft.
- Wael Maktouf analyzed the data, authored or reviewed drafts of the article, and approved the final draft.

### Human Ethics
The following information was supplied relating to ethical approvals (*i.e.*, approving body and any reference numbers):

The Ethics Committee of South Ethics Committee for the Protection. Persons of Tunisia (No. 0477/2022).

### Data Availability
The raw measurements are available in the Supplemental File.

### Supplemental Information
Supplemental information for this article can be found online at http://dx.doi.org/10.7717/peerj.16428#supplemental-information.

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
