# Peer review of "The impact of obesity on static and proactive balance and gait patterns in sarcopenic older adults: an analytical cross-sectional investigation"

_PeerJ, doi:10.7717/peerj.16428_

## Round 0.1 · original submission · Major Revisions

· Academic Editor

Major Revisions

Dear Authors,
The reviewers and I have completed our evaluation of your manuscript and recommend a major revision before re-submission.

Please review the comments and resubmit your revised manuscript.

·

Basic reporting

The paper was well written. However, The following points need to be corrected.


Line 137: Please add the references of the Ricci and Gagnon scales and the MoCA test.

Line 153: Please describe the impedance meter in detail

Line 156: Please add the references to this test. Why did the authors select the Romberg Test as a static balance of SO? Any prior studies showing the usefulness of the Romberg test in obesity or weakness should be cited.

Line 163: Same as line 156. Please demonstrate the usefulness of TUG and functional reach by citing prior research.

Table 1: Please add the unit of body mass.

Experimental design

The study was well-designed. Please respond to the following comment.

Line 128. Why did the authors use an effect size of r = 0.35 to calculate the sample size?

Validity of the findings

no comment.

Additional comments

no comment.

Reviewer 2 ·

Basic reporting

Important point: You talk about sarcopenic older people, but you did not describe whether they were indeed sarcopenic. Did you really only recruit sarcopenic people, and if so, what was the definition you used? I think you just compared old people without to those with obesity.
The title should not so much be ‘The interplay’ but rather ‘The impact of obesity on…’.
Abstract
I suggest to rephrase ‘to discern obesity’s role in shaping…’ into ‘the impact of obesity on..’ as it is not a factor that regulates the ‘shape’ of balance, but rather it may have an impact on balance. In this objective you also use the abbreviation ‘SO’ without introducing it. Please make sure you introduce each abbreviation at first use.
In the objective section I also suggest to replace ‘anthropometric indicators’ with ‘anthropometric characteristics’
In the Results of the abstract I don’t think the TUG time achieved was higher in the CG than the SO group., but rather the opposite. I know what you mean, that is that they performed better, but rephrase to reflect this. Check this throughout the manuscript!
When talking about the duration of the phases in the gait cycle, perhaps talk about it as being ‘longer/shorter’ rather than ‘higher/lower’. Incidentally, it is ‘swing’ rather than ‘wing’ phase.
Concerning the correlation with TUG: that was a negative correlation, where an increase in LBM resulted in a shorter time to complete the TUG, reflecting that the performance was better the lower the LBM. However, I have to issues here: 1) LBM per se may just differ between people because of differences in stature. I therefore rather see correlation with %LBM that you also calculated (incidentally, when trying it in your data it seems the correlation is even stronger); 2) plot the data, and you will see that there is no correlation in the CG, but there is a correlation in the SO, and I think you can then see that there might be a cut-off above which the TUG increases (poorer performance) (incidentally, some papers have suggested that if the muscle force per body mass is below a certain limit indeed TUG performance decreases with decreasing muscle force to body mass ratio), while above that threshold there is no relationship with . Consider this also for all the other correlations to separate the 2 groups and to make sure you consider differences in stature. I think, as %LBM + %Fat = 100% (or close to it), there is no need to show the correlations with %fat if you show the correlations with %LBM. As seen in the abstract, they are, as expected, the inverse of each other. Just choose 1 of them.
The first sentence of the conclusion is incomplete.
Many of these comments need to be considered throughout the manuscript.
METHODS
Line 105: Recruitment took place between 1-4 weeks? I guess recruitment took place over a period of 4 weeks.
Line 153-154: I don’t think calculating the actual lean and fat mass does not add to the info you need.
Line 183 is a repeat of line 176. In both sentences make sure the ‘th’ is superscript.
In the statistics section, I do not understand why you use an ANOVA and not just a t-test, as there are only 2 groups. I also don’t understand what the Bonferroni correction has to do here?
RESULTS
The first paragraph could just be summarised as ‘Body mass, BMI, body fat% and FBM were all higher in the SO than CG group (p<0.001).’ Apply this throughout the Results.
You write that there was no significant difference between groups in LBM, but I suspect %LBM was lower in the SO than CG. Can you add this information?
IN the second paragraph the order of the discussion of the parameters is not the same as the appearance in the table. It would be helpful to match the sequence.
Line 212: the time achieved in the TUG was LOWER/LESS rather than higher in the CG. A recurring error also seen in the abstract!
Table 4: Add to some cases the ‘0’ before the ‘,’. Also, in English the ‘,’ should be replaced with a ‘.’.
It would be illustrative to show the correlations for each group separately also. I think the correlations with FBM and LBM are not the best, as they don’t consider the fact that people may just have more LBM and FBM due to being taller. I therefore suggest to just stick to the correlation with body mass, BMI and % body fat, and show them in figures with different symbols for the CG and SO. I expect you will find something interesting!

DISCUSSION
Line 246-247: You write there were associations with obesity, but you don’t say what associations. I think it is better to be specific so the reader knows what obesity does.
Line 247-248: It is rather odd that ‘the analysis of walking dynamics accentuates the considerable influence of obesity’. If indeed the analysis does accentuate it, then don’t do the analysis (I hope you get what I mean). I think you mean that this analysis revealed associations (say what associations) of walking parameters with obesity.
Line 249: You say a strong link was found, again without specifying what the link is. To clarify my point, I can tell you ‘My brother and I differ in height.’, but this is not really informative. It helps the reader much more with as many words when I tell ‘My brother is taller than I am’. Consider this throughout the Discussion.
Line 254-255 and elsewhere: replace ‘equilibrium’ with ‘balance’. Also, throughout the manuscript don’t talk about a ‘decline’/’reduction’/’elongation’ in OS compared in CG, as a decline suggests a process, where here you only observed that values in OS were lower than those in CG.
Line 290: It is said here that step length was larger in SO than CG, while in Table 3 I see the opposite. Please adjust.
Note that it is interesting that there is no significant correlation between handgrip strength and gait speed, TUG and reach casting doubt on the often-used handgrip strength as a marker of functional capacity. Perhaps important to mention.

Experimental design

See above: appropriate design.

Validity of the findings

Findings appear fine, but some attention need to be given to the presentation of TUG data. I have also some other suggestions indicated above that may well reveal some additional interesting things.

---

## Round 0.2 · Minor Revisions

· Academic Editor

Minor Revisions

Please revise the manuscript according to the reports received from the reviewers

·

Basic reporting

The authors responded appropriately to the initial peer review comments.

Experimental design

no comment.

Validity of the findings

no comment.

Additional comments

no comment.

Reviewer 2 ·

Basic reporting

Well written

Experimental design

appropriate

Validity of the findings

Good, but I am puzzled that in the response the authors sya there is no relationship with %lean body mass in the control old people, but there is in the sarcopenic old people. Figure 2 suggests otherwise, where als the so-called sarcopenic old people have a higher, rather than a lower %lean body mass!!! There is thus something wrong in these graphs; I suspect the symbols were mixed up, at least in 2 of the 4 panels.

Additional comments

The figure needs to be checked carefully.

---

## Round 0.3 · accepted · Accept

· Academic Editor

Accept

Your manuscript has been accepted for publication. Congratulations!